# Biopigments of Microbial Origin and Their Application in the Cosmetic Industry

**Manal Jameel Kiki** 

Department of Biology, College of Science, The University of Jeddah, Jeddah 23218, Saudi Arabia; mjkiki@uj.edu.sa

**Abstract:** Along with serving as a source of color, many microbial pigments have gained attention as interesting bioactive molecules with potential health advantages. These pigments have several applications in the food, agrochemical, medicine, and cosmetic industries. They have attracted the attention of these industries due to their high production value, low cost, stability, and biodegradability. Recently, many consumers worldwide have noted the impact of synthetic dyes; thus, natural pigments are more in demand than synthetic colors. On the other hand, the cosmetic industry has been moving toward greener manufacturing, from the formulation to the packaging material. Microbial pigments have several applications in the field of cosmetics due to their photoprotection, antioxidant, and antiaging properties, including inhibiting melanogenesis and acting as natural colorants for cosmetics, as some microorganisms are rich in pigments. More investigations are required to estimate the safety and efficacy of employing microbial pigments in cosmetic products. Furthermore, it is necessary to obtain information about DNA sequencing, metabolic pathways, and genetic engineering. In addition, unique habitats should be explored for novel pigments and new producing strains. Thus, new microbial pigments could be of consideration to the cosmetic industry, as they are ideal for future cosmetics with positive health effects.

**Keywords:** natural pigments; biopigments; microbial pigments; cosmetics; cosmeceuticals; biological activity; photoprotection; antioxidants; sunscreen

## 1. Introduction

The expansion of civilization led to the widespread use of coloring. Pigments have several uses in a wide range of industries, including agriculture, textiles, medicines, food, and cosmetics, and have become a necessary component of our everyday lives [1–3]. Natural pigments were the sole source of color available until synthetic pigments became common. They have been used for a variety of applications, including coloring leather, fur, and natural fibers such as wool, cotton, and silk. Additionally, they are used to create inks, watercolors, and artist's paints, as well as to color cosmetic products [4,5]. The usage of natural colors has been reduced because of the availability of convenient and inexpensive synthetic pigments since Perkin first introduced synthetic dyes in 1856. It is commonly known that certain synthetic dyes' production is prohibited owing to the product's carcinogenicity and the impact of disposing of their industrial waste in the environment [5,6].

Microorganisms are a rich source of new bioactive chemicals because they give rise to high yields and productivity [7]. Natural pigments are among the bioactive substances of microbial origin that have attracted the interest of the industry due to the increasing need for the development of novel, degradable, safe, and environmentally friendly products [8]. The industry can produce microbial pigments for use in a variety of products, including food and cosmetics. In addition to enhancing the marketability of products, natural pigments exhibit beneficial biological properties such as antimicrobial and antioxidant properties. Therefore, it is crucial to investigate the numerous natural

sources of colorants and their uses [5]. Due to the usage of various colorings in cosmetics and its global market, efforts have been made to investigate the use of biopigments in cosmetics, particularly in skincare products [9]. The skin is an important organ in the body, since it protects us in many different ways. As people grow older, their skin becomes thinner and loses its elasticity and moisture capacity [10,11]. Because the skin is exposed to chemicals and ultraviolet (UV) radiation, it might lose its antioxidant capability and age faster [11]. Therefore, skincare is not only important for the skin's appearance and health, but also for the skin's barrier function [12].

Many cosmetic products are composed of synthetic chemicals, which may cause side effects in the body; for example, some pigments may cause damage to cells and some UV filters may even cause tumor formation [13]. Moreover, metals are widely used in cosmetics as pigments—for example, in eye shadow, lipstick, blush, and eyeliner. However, some metals, such as cadmium and chromium, are harmful to the human body and may even affect human metabolism. However, the toxicity of cosmetics was not taken into consideration in the early period, which led to a variety of negative adverse effects, including deformities, blindness, and even death. Although there are limitations to their use, these limitations are not entirely effective. The cosmetics that are absorbed by our body may act as carcinogens, reproductive toxins, endocrine disruptors, mutagens, and sources of neurotoxicity [14].

Consumers who are concerned about safety are prepared to spend more on cosmetics that contain natural components that are better for their skin [13]. If the products are colored with natural substances, they have a high market value. Natural pigments can boost a product's marketability while also exhibiting beneficial biological properties, including antioxidant and antimicrobial properties. Consequently, it is crucial to investigate the numerous natural sources of colorants and their uses [5]. By 2027, it is predicted that the global market for natural cosmetics will reach USD 54.5 billion [15]. With the increasing consumer demand and the expansion of the cosmetic industries, it is required to produce many active natural pigments. Otherwise, only minor pigments, such as astaxanthin, monascus, and carotene, are commercially available [16].

Researchers are now exploring novel strains, new natural pigments, extraction methods, and pigment applications. In this regard, this review highlights the characteristics, the classification, and the application and significance of microbial pigments from prokaryotic and eukaryotic sources and their application in the cosmetic industry.

## 2. Natural Pigments and Artificial Colors

Synthetic dyes and natural pigments are utilized extensively in several industries, including textiles, paper, food, agriculture, and cosmetics. Green technology advocates for natural, low-toxicity materials in modern manufacturing lines [5]. To meet the increased demand for the enhanced color of various items, artificial dyes have been manufactured on a huge scale. They still outweigh the advantages of natural dyes in terms of large-scale manufacturing at consistent quality, a low cost, and a wide range of color variations [3].

In general, synthetic dyes are composed of toxic chemical substances such as chromium, lead, mercury, copper, toluene, and benzene, which are harmful to health. Several synthetic colorants originally approved for use in pharmaceuticals, food, and cosmetics research by the Food and Drug Administration (FDA) were later shown to be carcinogenic [17]. However, several studies prove that synthetic colors are harmful to human and environmental health. In addition, the waste by-product is a significant difficulty since synthetic dyes are non-biodegradable, non-renewable, and carcinogenic and produce hazardous waste contamination [3,18].

Biological pigments have been investigated for a long time, but because of the possible health and environmental risks associated with synthetic dyes, they have attracted increased interest from the industry. Biological pigments derived from biological sources have a specific color that corresponds to their structure. Animals, plants, and microorganisms all contain them [19]. Microorganisms such as bacteria, fungi, and microalgae provide

an alternative supply of natural colors [17]. As a source of pigment, microorganisms are better than plants because they grow quickly in a cheap medium, are easy to process, and can be grown all year round, regardless of the season. In addition, the large-scale usage of plants might harm rare species; hence, the practice is not sustainable [17]. Moreover, pigments generated by microbes are more than simply colors; they also contain a variety of chemical components with multiple biological functions [20].

## 3. Pigments Produced by Microorganisms

Microbial pigments are biological colors derived from microbial cells. In addition to serving as colorants, several microbial pigments are employed to benefit human health by supplying essential nutrients or chemicals. Additionally, some have biological effects, including antimicrobial, anticancer, immunosuppressive, and anti-inflammatory effects [21]. Microbial pigments are utilized for a variety of purposes based on their color and biological activity; for example, fluorescent microbial pigments are employed in labs to mark antibodies [9]. Some pigments can also be used to measure pH changes or to show the progress of specific processes by changing their color [22]. Microbial pigments are in high demand for use in the cosmetic, textile, food, and pharmaceutical sectors. The high demands for microbial pigments are due to their high production value, easy cultivation, low cost, availability, stability, biodegradability, eco-friendly nature, unlimited resources, and multiple applications in several disciplines, such as biomedical, agriculture, evolutionary, ecological, and industrial fields [18,23].

It is well known that many microorganisms produce a wide range of pigments with several biological features and various commercial uses [16]. Furthermore, microbial pigments are significant in the molecular physiological processes of microbes because they serve as a mechanism of adaptation to varied severe conditions, have a protective role against solar radiation, and are engaged in functional activities such as photosynthesis [24]. Moreover, because microorganisms are affected by the environment, they produce a wide range of pigments with unique properties that are connected to the relationship between the microorganism and the ecosystem. Today, a remarkable variety of microbial pigments found in many habitats have been identified, including melanin, carotenoids, flavins, phenazines, quinones, monascines, and violaceins [25]. Numerous studies proved that these natural pigments have a range of advantageous features, such as pro-vitamin A and antitumor, in addition to desirable traits including thermal stability, pH stability, and photostability [26]. Pigments are abundantly produced by microorganisms such as fungi, yeasts, microalgae, and bacteria [8].

Pigments such as melanin, carotenoids, prodigiosin, violacein, pyocyanin, zeaxanthin, and actinorhodin are all produced by bacteria [27]. Many bacterial pigments have been shown to have UV protection and antioxidant characteristics, and many have shown potential biological uses, such as anticancer, antimicrobial, and antimalarial activity [17]. Based on this, bacterial pigments appear to hold promise for a variety of novel biotechnological and industrial applications [18]. However, due to their short life cycles, low susceptibility to climatic and seasonal variations, simplicity of scaling, ability to create pigments in a variety of colors, and ease of genetic manipulation, bacteria have several distinguishing advantages over their counterparts [8,24]. Recent advances in genetic engineering have enabled the modification of bacteria to manufacture the pigment of interest. For instance, *Streptomyces coelicolor*, which generates a blue pigment, may be genetically engineered to create orange, yellow, red (anthraquinones), or brilliant yellow (kalafungin) pigments [27]. Carotenoid pigments have been mentioned as well; they exhibit antioxidant properties and are safe for use as natural colorants in the food, pharmaceutical, textile, and cosmetics sectors [3,28]. In comparison to other bacterial groups, actinobacteria are more likely to produce pigments. The most prevalent pigments found in actinobacteria are melanin, with a color ranging from olive or brown to black; carotenoids that range from yellow, pink, and red to violet; and a blue color linked to actinorhodin [29].

It is well known that filamentous fungi can produce a variety of colors, including carotenoids, flavins, monascins, phenazines, quinones, and indigo, displaying a large color spectrum [3]. Fungal pigments have been linked to valuable biological activities, including antioxidant, antimicrobial, anticancer, immunological suppressor, and anti-inflammatory [30]. Due to their ability to absorb UV radiation, they have potential uses in the healthcare, food, cosmetics, and textile industries [3,31]. Additionally, several pigments have been identified and have been used for a long time as taxonomic markers; some of them are even accessible commercially for cell staining [8,32]. Pigment production has also been identified in yeast species such as *Sporidiobolus*, *Xanthophyllomyces*, *Rhodotorula*, and *Pichia*. According to reports, some of them generate large quantities of torulene, toru-larhodin, poly-hydroxy carotenoids, and β-carotene [3]. Yeasts are unique in that they create the red, orange, and yellow pigments that are employed in the pharmaceutical, aquaculture, and wine industries [8,33].

Microalgae such as *Nostoc*, *Spirulina*, *Chlorella*, *Dunaliella*, *Haematococcus*, and *Porphyridium* generate colors such as chlorophylls, carotenoids, and phycobiliproteins (PBPs), which are water-soluble, non-toxic proteins found mostly in Cyanobacteria, Rhodophyta, and Cryptophytes [34]. PBPs have been widely used in the pharmaceutical, cosmetics, and food industries due to their potent absorbance, antioxidant properties, and fluorescence characteristics [35]. Microalgal pigments, in general, have high commercial value as natural colorants in the cosmetic, pharmaceutical, and nutraceutical industries, as well as applications in clinical research in molecular biology, and as natural colorants in the food, painting, textile, and poultry feed additive industries [3,36].

## 4. Classification of Microbial Pigments

There is a wide spectrum of color seen in microbes, including cream, yellow, orange, pink, red, purple, green, grey, brown, black, blue, indigo, and metallic green. These pigments can be categorized according to their chemical, visual, and spectral properties, as well as their origin [16,18]. Pigments of microorganisms are widely divided into phycobiliproteins, carotenoids, melanin, prodigiosin, violacein, and rhodopsins [37]. Most microbes have innate pigments; some non-pigmented microorganisms acquire pigment features from pigmented ones. As a result, microbial pigments are divided into two categories: inherent and acquired pigments. Pigmented microorganisms frequently produce diffused or non-diffused pigments in a culture medium. However, some colors are insoluble in water—for example, indigoidine (blue) [38], red pigments [39], and violacein [40].

Commercially, natural pigments such as chlorophylls and carotenoids are employed to create hues of yellow, orange, red, and green [41]. However, for blue colors, there are only a few natural pigments; thus, there is significant corporate interest in discovering new sources. Significant supplies of blue natural pigment would increase the range of natural colors used in industrial applications, providing an essential color that could be utilized alone or in combination with current natural pigments. Evaluating studies on blue pigments presents several challenges, since chemical databases often classify compounds based on their composition and molecular weight rather than their color [42].

It is well known that the blue color is uncommon in nature due to the complicated and sporadic electrical structures needed to absorb photons around 560–700 nm [42]. Animals' blue hues are often caused by nanoscale structures reflecting blue light, rather than by the presence of true-blue pigments. For example, the wings of the *Morpho* sp. butterfly and the feathers of the *Sialia* sp. bird are both blue [43]. Moreover, *Streptomyces coelicolor* has been extensively investigated as a producer of actinorhodin, a blue pigment. This antibiotic-active anthraquinone compound has been proposed for usage in cosmetics and foods. It has low acute toxicity and strong stability under light and heat. However, anthraquinones, such as actinorhodin, often do not exhibit a blue color at an acidic pH. The blue color of the molecule appears only at pH levels greater than 8, which has restricted its application in food and beverages [43].

### 4.1. Carotenoids

Carotenoids are liposoluble organic pigments found naturally in photosynthetic organisms and plants, and they are responsible for most of the red, orange, and yellow colors in nature [44]. There are around 600 known carotenoids, which are all tetraterpenoids and are classified into two groups: xanthophylls (oxygen-containing) and carotenes (which are contain no oxygen and are purely hydrocarbons). The most significant carotenoids are lycopene, lutein, alpha- and beta-carotenes, cryptoxanthin, neoxanthin, violaxanthin, and zeaxanthin [27,45]. Microorganisms that produce carotenoids are numerous and isolated from many terrestrial and marine habitats. The most important function of carotenoids is their ability to absorb light energy. They absorb light and transfer the excitation energy to chlorophyll, increasing the wavelength range of the collected light [45]. They shield chlorophyll from light damage. Moreover, they are utilized as vitamin supplements and are crucial in oxidative stress defense. Their consumption helps to avoid sunburn and photoaging [46]. Additionally, epidemiological studies have shown a decreased incidence of lung cancer in those with high β-carotene consumption [47].

Carotenoids are commercially employed as food colorants, animal feed additives, and in obesity therapy. They are now used in the medicinal, cosmetic, and nutraceutical industries [27]. The stability in emulsions, solubility, antioxidant properties, and capabilities of the carotenoids derived from microorganisms encourage their use in cosmetics, as they are desirable for use in sunscreens and other anti-UV products [48]. Furthermore, lycopene, β -carotene, and astaxanthin are among the carotenoids that have been reported and employed as photoprotection agents owing to their antioxidant effect, which fights against free radicals caused by UV radiation to protect the skin from erythema [49].

### 4.2. Melanin

Melanin represents a class of diverse complex natural pigments produced by all organisms, from bacteria to humans. Chemically, it is the result of the polymerization of phenolic and indole rings, with tyrosine being the primary precursor [44,50]. Because there are so many different sources of melanin, it has a very varied and heterogeneous structure, which affects its composition, function, size, and color. In addition, the physical features of melanin (large molecular weight, hydrophobic nature, and highly negative charge) make it difficult to detect its structure via analytical methods. It is also insoluble in most solvents and resistant to chemical degradation [51]. It provides a variety of biological roles, including antimicrobial activity, photoprotection, antioxidant effects, and thermoregulation. Melanin is the most visible pigment in humans, and it is responsible for the color of our hair, skin, and eyes [51]. Plants and insects use melanin to build their cell walls and straighten their cuticles, respectively. In addition, melanin synthesis has been associated with resistance to visible and UV light, protection against reducing and oxidizing chemicals, survival and competition in harsh environments, and resistance to cell-wall enzymes [44,52]. Melanin is often utilized in pharmacology, agriculture, and medicine, and has significant applications in cosmetics. It is also used in the production of monoclonal antibodies for the treatment of human metastatic melanoma [27,53].

### 4.3. Phycobiliproteins

Phycobiliproteins (phycoerythrins, phycocyanins, and allophycocyanin, which are red, blue, and green, respectively) are made up of α and β subunits that form covalent bonds between phycobilins and cysteine residues. In photosynthetic organisms, there are many primary phycobilins, but the two main ones are phycoerythrobilin and phycocyanobilin. These colored proteins primarily function as secondary photosynthetic pigments and have antioxidant properties. This protein pigment functions as a photosynthetic light-harvesting molecule in addition to chlorophylls [54,55].

### 4.4. Prodigiosin

A red pigment called prodigiosin was initially discovered in *Serratia marcescens*. Additionally, *Vibrio psychroerythrous*, *Streptoverticillium rubrireticuli*, *Pseudomonas magneslorubra*, and *Alteromonas rubra* have all been observed to produce prodigiosin [56]. Microbes that can produce prodigiosin have been found in many marine environments [57,58]. Prodigiosin is a very effective medicinal chemical, particularly as an immunosuppressant and anticancer drug. It also has insecticidal, antifungal, antibacterial, and anti-malarial properties [27].

### 4.5. Indigoidine

Indigoidine is a water-soluble blue pigment generated by a few microorganisms, such as *Streptomyces chromofuscus*, *Vogesella indigofera*, and *Arthrobacter* isolates from the Antarctic [44]. Indigoidine's physiological function is unclear; however, it has recently been revealed that it may give resistance to oxidative stress, as well as protection via its antimicrobial action. Thus, microorganisms that produce indigoidine may have a competitive feature in their natural habitats, owing to its antioxidant and antimicrobial capabilities. Moreover, it has a role as a motility-related intracellular signaling molecule [44,59].

### 4.6. Violacein

Violacein is a purple pigment derived from bis-indole with a few unique biological properties [60]. This pigment can be produced by *Pseudoalteromonas*, *Alteromonas*, *Collimonas*, and *Chromobacterium* spp. Violacein displays a peak in UV absorbance (260 nm), which suggests a possible function in UV and visible radiation protection. In addition to UV resistance, violacein possesses antimicrobial action against Gram-negative and -positive bacteria, as well as fungi [44,61].

## 5. Application of Microbial Pigments in Cosmetics

Before the idea of cosmetics as we know them today, people used natural ingredients to enhance their appearance and improve their health. Thus, mineral ingredients, herbal pastes, and oils were used to produce cosmetics for a long time [62]. Cosmetics are products designed to enhance the morphology, structure, and look of the skin using active substances that are suitable for various skin types [63]. Recently, the cosmetic industry has moved toward greener production, from the ingredients to the packaging. The use of cosmetics for health purposes introduced a new aspect of the industry called cosmeceuticals, a new sector of this vibrant business domain that is currently searching for new prospects besides beauty. Although "natural" does not always imply "healthy", it is true that the substances utilized in natural cosmetics are possibly valuable to health [62].

Microbial pigments offer exceptional antioxidant and antiaging effects, making them ideal for use in cosmetics. In addition, cosmetic products may also benefit from the vibrant colors of the ingredients. Several cosmetics using these colors are now on the market; however, many of them do not attract attention owing to safety requirements. The antioxidant, melanogenesis-inhibiting, and photoprotective properties of these pigments make them excellent for future cosmetics with health advantages. Figure 1 illustrates the various applications of biopigments derived from microbial origin in cosmetics.

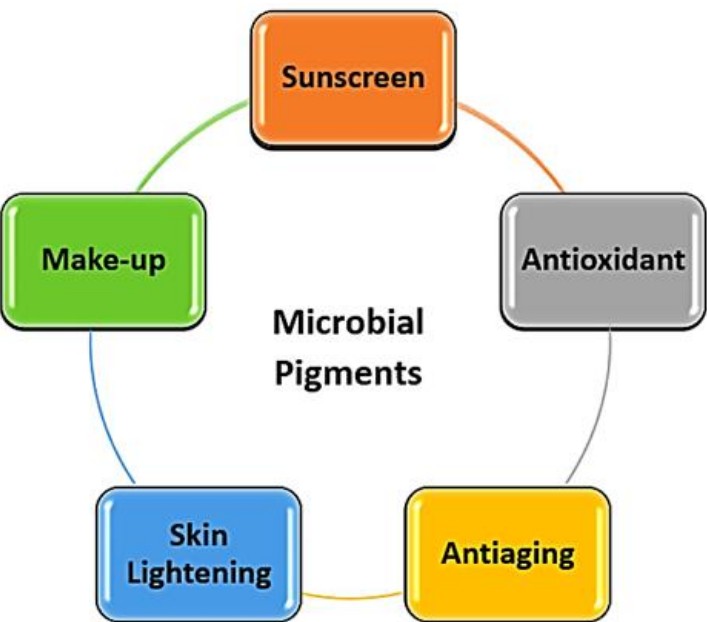

**Figure 1.** Application of microbial pigments in cosmetic products.

*5.1. Photoprotection and Antioxidant Capacity*

Natural antioxidant agents present in skin may inhibit reactive oxygen species (ROS) and prevent cell instability. However, when the level of ROS is raised by UV radiation, these defenses may be overcome. The resultant oxidative stress may cause DNA deterioration and cell damage due to free radicals, especially to lipid and protein membranes. Additionally, ROS may cause cell death via necrotic or apoptotic processes, and the development of wrinkles and dry skin makes this obvious. Photoaging consequences such as skin cancer, melanoma, and cutaneous inflammation may therefore be caused by ROS build-up [10]. Furthermore, chemical UV filters may not provide complete skin protection since they absorb rather than reflect all UV rays that reach the skin. Some sunscreens irritate the skin, trigger allergic responses, and lack light stability. As a result, antioxidant molecules must be included in sunscreens, particularly topical treatments that reduce UV damage to the skin [49].

The cosmetic industry is now searching for innovative, eco-friendly, UV-resistant compounds [13]. It is interesting to note that the cosmetic industry may greatly benefit from the antioxidizing capability of colorants produced by microorganisms such as microalgae and cyanobacteria, since they can be employed as both natural colorants and antioxidants. Specifically, this consists mostly of photosynthetic pigments such as carotenoids [63]. Carotenoids have captured the attention of the cosmetic industry because they are significant natural antioxidants that reduce the production of free radicals and, as a result, lessen skin photodamage [64]. They also offer potential anti-UV properties, which encourages the development of photoprotective cosmetic products that aim to minimize the use of chemical filters or replace them with natural sunscreen ingredients [49].

Lycopene, astaxanthin, and β-carotene are among the most often characterized carotenoids and are used as photo-protectants due to their powerful antioxidant activities [65]. For example, β-carotene, a precursor to provitamin A and a scavenger of free radicals, is directly linked to its accumulation in the skin after intake and subsequent conversion to vitamin A in the body, which aids in the synthesis of melanin. Additionally, lycopene is a strong natural antioxidant that may be added to sunscreen since it can neutralize oxygen free radicals [13,63]. It is an effective antioxidant and significantly affects skin roughness. This implies that increasing the skin's antioxidant levels might significantly decrease skin roughness [66].

Astaxanthin is also thought to be nature's strongest antioxidant; it is a powerful antioxidant that protects against oxidative stress and lipid peroxidation. This feature

has also been linked to protection from wrinkles and spots [49,66]. Astaxanthin is also important in human metabolism for skin protection against UV-induced photooxidation; therefore, it may be employed in natural sunscreen cosmetics [63]. Astaxanthin has ten-times the antioxidant activity of zeaxanthin, lutein, and β-carotene. This is due to the carbonyl functional groups in astaxanthin's ionone ring [67]. Astaxanthin has been used as an active ingredient in certain cosmetic brands across the world due to its significant antioxidant activity [13,68].

Studies using cosmetic formulations containing these molecules show promising outcomes in terms of oral intake, supporting the concept of employing topical carotenoids to counteract oxidative stress caused by UV radiation [66,69]. However, carotenoids derived from microorganisms have shown increased use in the ingredients of sunscreens and anti-UV treatments due to their potent antioxidant capabilities, oil solubility, and stability [49].

Otherwise, a melanin-infused cream (designated as cream F3) was produced by combining concentrations of melanin from a bacterium (*Halomonas venusta*) with an extract of seaweed (*Gelidium spinosum*). Cream F3 displayed photoprotective action and high SPF values, as well as excellent wound-healing efficacy. Furthermore, the cream demonstrated antibacterial action against skin pathogens *Staphylococcus aureus* and *Streptococcus pyogenes* [70,71]. However, there are many other studies that have focused on photoprotection activity of microbial pigments, as shown in Table 1.

**Table 1.** Some studies focused on the photoprotection activity of different microbial pigments.

| Pigments | Origin | Producer Organism | References |
|---|---|---|---|
| Carotenoids Yellow pigment | Bacteria | *Flavobacterium* sp. | [72] |
| Carotenoids Orange pigment | Bacteria | *Brevibacterium* sp. | [72] |
| Melanin | Bacteria | *Vibrio natriegens* | [10] |
| Carotenoids Lycopene | Cyanobacteria | *Anabena* sp. | [73] |
| Carotenoids Scytonemin | Cyanobacteria | *Nostoc* sp. | [74] |
| Biopterin | Cyanobacteria | *Oscillatoria* sp. | [75] |
| Phycocyanobilin | Cyanobacteria | *Spirulina* sp. | [76] |
| Carotenoids Astaxanthin | Chlorophyta | *Haematococcus lacustris* | [77] |
| Phycoerythrobilins | Rhodophyta | *Porphyridium* sp. | [76] |
| Phycobiliproteins | Rhodophyta | *Porphyridium aerogineum* | [78] |

*5.2. Antiaging Compounds and Skincare Products*

The threat of stress and pollution has been imposed on human life due to urbanization and globalization. Stress and pollution have led to the issue of early skin aging [79]. The effects of aging on the skin include thinning, wrinkles, fine lines, enlarged pores, fragility, laxity, and dryness. Intrinsic aging refers to skin deterioration in the form of a loss of elasticity, increased transparency, and vascular prominence. It is represented in the variations in skin tone that may be noticed in young adults, newborns, and elderly people [10]. On the other hand, extrinsic aging, is caused by exposure to UV radiation and the associated irritants. It is characterized by a decline in the keratinocyte dysplasia in the epidermis and dermal extracellular matrix, which leads to pigmentation, laxity, wrinkles, and coarseness [20]. There are three types of UV light: (A, B, C). UVA may pass through the dermis and cause symptoms that affect the skin, in addition to causing wrinkles. UVB

is more likely to cause cancer and causes the skin to turn red and burn. On the other hand, the ozone layer blocks UVC, so it does not harm the skin [10,80].

Antioxidants and antiaging creams have been the most popular solutions for this and are in constant demand as individuals wish to appear more attractive and healthier. With increased concern about biological safety and toxicity, consumers are turning to natural products, which are safer to use [79]. The present demand for sustainable ingredients in the cosmetic industry has boosted the search for new active natural ingredients. For example, carotenoids are some of the most important microbial pigments that are safe and have been used in different ways in the cosmetic industry because of their color [81]. The biological functions of carotenoids in the skin are linked to both their role as a precursor to vitamin A and their ability to serve as a UV photo-protectant. As a result, the mechanism of action in preventing skin damage caused by the sun is the direct absorption of UV rays. Some clinical investigations have shown that employing carotenoids as nutricosmetic components may help to prevent premature skin aging [82].

It has been seen that ß-cryptoxanthin, a carotenoid present in *Dunaliella salina* and other microalgae, can stimulate the production of hyaluronic acid, which is a glycosaminoglycan important in skin hydration [63,83]. In addition, the cyanobacterium *Anabaena vaginicola* exhibited a much greater lycopene concentration than any other natural source [73]. Lycopene is already employed as an antiaging ingredient in personal care products; therefore, cyanobacteria and microalgae may provide lycopene for cosmetic purposes [63]. Furthermore, astaxanthin produced from *Haematococcus pluvialis* improved the skin texture, wrinkles, corneocyte layer, and moisture content [84]. There are many commercial products that contain this dye, such as AstaBlanc, a wrinkle- and spot-fighting product formulated with astaxanthin, which has been sold by Japanese cosmetics company Kose. Moreover, the Swedish company AstaReal AB has marketed the natural astaxanthin (AstaReal), which revitalizes photodamaged skin, eliminates wrinkles, and improves skin elasticity [85].

In a previous study, lutein and zeaxanthin were given to three groups of volunteers: the first group received both molecules orally and topically, the second group received the therapy topically, and the third group only took them orally. The findings demonstrated that all treatments significantly improved cutaneous hydration and skin elasticity in the combination therapy rather than in the standalone therapies. The best level of antioxidant protection was also achieved by the combined effects of oral and topical therapies [86]. In another study, the potential of phycobiliproteins and carotenoids from the cyanobacterium *Cyanobium* sp. was evaluated, and the results revealed that phycobiliprotein extracts have anti-hyaluronidase and anti-collagenase activity, whereas the carotenoid extract showed anti-hyaluronidase activity. Consequently, both extracts have the potential to be promising natural antiaging ingredients owing to their bioactive capability, making them natural, viable, and sustainable compounds for cosmetic purposes with very easy storage and formulation [62].

### 5.3. Whitening Effect

The epidermis and dermis of the skin provide protection for the human organism. The epidermis, which is the skin's top layer, is mostly composed of melanocytes and keratinocytes. Melanocytes transmit melanin to keratinocytes, allowing the skin to create melanin caps and decreasing the UV-induced DNA damage of the epidermis [13,87]. The quantity, distribution, and type of melanin in the skin are the primary factors in determining skin color [88]. Tyrosinase is a key enzyme in the formation of melanin [89]. Tyrosine may be converted into dopaquinone by its catalysis, which is then transformed into melanin by several chemical reactions [13,90].

Skin whitening is a popular cosmetic procedure around the world, especially in Asia [91]. This is because, in Asian culture, white skin is considered attractive. As a result, skin whitening products represent a significant portion of the cosmeceutical industry in this area, with considerable growth projected in the future. Sun exposure boosts both

tyrosinase and melanosome production. Melanosomes develop to generate melanin, which migrates to keratinocytes and degrades to promote tanning and skin melanization. The tan may then be removed by the loss of melanin caused by desquamation. The most frequent method for skin whitening is to utilize tyrosinase inhibitors, since the enzyme catalyzes the pigmentation process [10,92]. In the cosmetic and pharmaceutical sectors, tyrosinase inhibitors are gaining importance owing to their ability to prevent pigmentation issues. Tyrosinase inhibitors are used in cosmetics for hyperpigmentation, particularly the development of freckles, and may cause a decrease in melanin production [93]. Tyrosinase and its inhibitors could also be used to produce medicines to treat albinism and piebaldism, which are both caused by a lack of pigmentation [53,94].

Despite the enormous number of in vitro tyrosinase inhibitors discovered, only a few have been demonstrated to have substantial therapeutic effects. Current research on the discovery of natural tyrosinase inhibitors has focused on marine algae. For example, it has been shown that fucoxanthin extracted from *Laminaria japonica* inhibits tyrosinase activity in UVB-irradiated guinea pigs and melanogenesis in UVB-irradiated mice. Additionally, oral fucoxanthin intake reduced the expression of skin mRNA associated with melanogenesis, indicating that fucoxanthin adversely affects melanogenesis at the transcriptional level [10,95]. Furthermore, the primary carotenoid in *Haematococcus pluvialis*, astaxanthin, has superoxide dismutase and catalase enzyme capabilities that protect human lymphocyte proteins and critical lipids from oxidative damage. It is a superior antioxidant to vitamins E, C, and other carotenoids [96]. According to studies, astaxanthin may be used topically and orally to reduce skin hyperpigmentation, stop the production of melanin, and enhance the health of the skin [10,84]. Moreover, marine-yeast-produced astaxanthin has been demonstrated to prevent age spots [97]. Many marine-organism-produced compounds that suppress tyrosinase activity have been utilized commercially, while others (hydroquinone) have been prohibited in all European nations because they pose a health risk to humans [98].

### 5.4. Pigment Substances

Cosmetics such as nail polish, lipstick, and others have attractive colors. Most industrial pigments are made from benzene, toluene, and other chemical reagents. Due to this, many people find them dangerous and unacceptable. In contrast, natural colors are used widely in some products because they are safe and stable [99]. Numerous investigations on microorganisms as sources of economical, stable, novel, and safe biological pigments have been reported [13]. Furthermore, certain invertebrates in water, such as corals and sponges, have exceptional colors, which may be connected to the photosynthetic pigments of symbiotic microorganisms [63]. Marine microorganism pigments are now used extensively in the cosmetic industry [13]. There is also interest in using certain pigments in the preparation of cosmetics. For example, eye shadow may be produced using phycocyanins, which are generated by thermophilic blue–green algae [100]. Additionally, the Japanese company Ink has been using phycocyanin extracted from *Spirulina* sp. as a coloring agent in eye makeup [101]. Additionally, natural pigments derived from red microalgae may be used to create pink and purple cosmetic colors [13]. For example, lipstick and eye shadow were developed in the form of creams or powders with pigments extracted from different red microalgae [85].

Biotechnologically, phycoerythrin has been utilized as a natural pigment in colored creams and cosmetics, and the phycoerythrobilins from *Spirulina* (Cyanobacteria) and *Porphyridium* (Rhodophyta) may be used in lipstick and eyeliner [12,76]. Moreover, phycobiliproteins derived from *Porphyridium aerogineum* are utilized as cosmetic colorants; this pigment is not affected by pH (4, 5) and its color is stable under light [63,78]. Phycocyanin has been examined for its antioxidant, immune-boosting, and anti-inflammatory properties. Additionally, phycocyanin's stability makes it a significant component in cosmetic products including eyeliners, eye shadow powders, and lipsticks [102].

Further, it was disclosed in a patent (US6740313 B2) that ankaflavin, one of the principal pigments derived from *Monascus* sp., was employed as a dermatological composition to generate a long-lasting skin pigment that resembles a natural tan. According to the formulation, this pigment immediately colored well in a skin test in vitro [103]. In another report, a bio lip balm prepared from a crude pigment extracted from *Streptomyces bellus* in a combination of shredded bee's wax, lanolin, and coconut oil suggested the use of melanin pigment as a major component in a variety of beauty products [104]. Furthermore, the strong green pigmentation of microalgae makes it easy to extract pigments such as chlorophyll for use in the cosmetic industry. In particular, chlorophyll from chlorophyte is employed as a colorant and in anti-inflammatory products [101]. Moreover, numerous extensive investigations have lately been gathered in reviews. For example, they show that chlorophyll has the potential to cover unpleasant smells, making it a useful component in deodorants, toothpastes, and other personal care products [63]. In addition, several ecological niches were surveyed for potential new supplies of microbial pigments that can be used in the beauty industry, and many pigment-producing strains were discovered, as illustrated in Table 2.

**Table 2.** Some biopigments of microbial origin isolated from new ecological niches.

| Pigment | Color | Organism | Isolation Area | References |
|---|---|---|---|---|
| Variecolorquinones | Yellow | *Aspergillus glaucus* | Mangrove roots | [105] |
| Aspergiolide | Red | *Aspergillus glaucus* | Mangrove roots | [105] |
| Bostrycin | Red | *Aspergillus* sp. | Coral reefs | [106] |
| Physcion | Yellow | *Eurotium cristatum* | Sponge | [106] |
| NIOM-02 | Red | *Penicillium* sp. | Marine sediments | [107] |
| 2,3-dihydrocitromycin | Yellow | *Penicillium bilaii* | Huon estuary | [108] |

## 6. Trends in Cosmetics and Pigments

Most biopigments, including carotenoids, are light sensitive, which restricts their usage and reduces the shelf life of the items that contain them. Additionally, most carotenoids have a unique color, which limits their usefulness in certain cosmetics [109]. Phytoene (PT) and phytofluene (PTF) are colorless carotenoids (CLCs) which exclusively absorb electromagnetic radiation in the UV range. They are a precursor in the biosynthetics of other colored carotenoids. CLCs are present in most carotenogenic organisms, including in microorganisms such as the microalgae *Dunaliella* sp. [110]. These molecules are part of a recently discovered type of carotenoid that has been applied in the cosmetics, food, and medicine industries. The benefits of colorless carotenoids over colored carotenoids for topical cosmetic usage are obvious: they allow for the use of effective amounts without staining the skin, which can occur with low levels of colored carotenoids [109].

Colorless carotenoids have mostly been used in wellness nutrition and cosmetics to prevent external and internal stress and to protect the body from oxidative stress. They also represent a required active component in topical formulations due to their anti-oxidative power, UV protection, and anti-inflammatory effects that can be achieved without color [109]. It has also observed that the CLCs enhance the formation of collagen and prevent the synthesis of collagenases [111]. In cosmetic formulations, CLCs may be found in moisturizers, sun protection products, and as enhancers of other biomolecules, such as CoQlO [109]. Skin care products containing CoQ10 claim that this molecule can energize the skin and slow the aging process. According to a previous study, topical skin care products containing CoQ10, and colorless carotenoids may provide improved defense against inflammation and early aging induced by sun exposure. This suggests that their prolonged usage may reduce the loss of collagen in the skin matrix during the normal aging process [112].

Concerning the cosmetic benefits of CLCs, some other studies have shown that topical products consisting of PT and PTF can help in skin lightening [111,113]. They have been used as skin-whitening agents in new cosmetics [109]. Aesthetically, this impact is signifi-

cant since darker colored skin is considered unattractive in many regions (especially in East Asia). This effect can be obtained by colorless PT and PTF and not by other carotenoids that stain the skin [110].

Recently, CLCs have gained a lot of interest due to their bioavailability and usefulness in nutrition, health protection, and cosmetics. However, they have not been as in depth as other carotenoids. Hence, more research is needed to discover new types of CLCs as well as their different uses, especially in the field of cosmetics, as one of the new trends in this area is the colorless cosmetics, which combine the effects of makeup with the benefits of skin care. Some cosmetic companies have started producing this type of distinctive cosmetics.

## 7. Conclusions and Perspectives

The cosmetic industry is growing at a fast rate. It is estimated that the value of natural cosmetics on the global market will reach USD 54.5 billion by 2027 [15]. Over the past few years, the rise in globalization has been a significant factor influencing the pigment industry. Even though artificial colors are more appealing and are often utilized on the global market, they are found to have many side effects, and some are not biodegradable, causing health and environmental problems. However, people all over the world have recently become more aware of the impacts of synthetic colorants. Hence, the demand for natural pigments derived from a variety of natural resources has surpassed that for artificial colors.

Natural microbial pigments offer several advantages over manufactured colors. There are several possible uses for microbial pigments, including textiles, food, medicine, and cosmetics. The variety of applications for microbial pigments in the cosmetic industry, including antioxidant, photoprotection, and antiaging functions, should stimulate interest and increase research into their efficacy as cosmeceutical and cosmetic components. While certain microorganisms are rich in pigments including melanin, carotenoids, and phycobiliproteins, the manufacture of natural colorants for cosmetics (such as eyeliners, eye shadows, lipsticks, etc.) might be another area of interest. Hence, the beauty sector may be interested in more thorough studies to discover novel colors from microbial cells.

It is crucial to understand that most microorganisms have a major advantage when it comes to producing their pigments on a wide scale because of their quick development on simple media. Hence, further knowledge of metabolic pathways, genetic engineering, and DNA sequencing to improve biopigment synthesis is one of the most promising strategies to overcome this challenge. As a result, these factors represent the increased relevance of discovering novel microbial pigments and uses. Moreover, exploring new and unknown niches may be helpful in discovering new sources of high-value pigments. Furthermore, the function of microbial pigments as cosmeceuticals will be further enhanced and established with the use of low-cost, environmentally friendly, commercially viable technologies and worldwide clinical trials.

To assess the effectiveness and safety of employing microbial pigments in the cosmetic industry, further research is still required. Moreover, the industry needs additional work to make many novel high-yield strains commercially viable. For example, blue pigments are uncommon in nature, and even manufactured blue colorants have stability challenges. Therefore, new sources of blue pigments need to be discovered, especially in terms of compounds of a natural origin, since they will have stronger commercial appeal. The future of microbial pigments will be more colorful and brighter if researchers improve purification and extraction methods to meet the rising demand of the global market. This includes looking for safe environmental methods, using less solvents and energy, and making them easy to scale up.

**Funding:** This research received no external funding.

**Institutional Review Board Statement:** Not applicable.

**Informed Consent Statement:** Not applicable.

**Data Availability Statement:** Not applicable.

**Conflicts of Interest:** The author declares no conflict of interest.

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
