# Peer review of "Biopigments of Microbial Origin and Their Application in the Cosmetic Industry"

_cosmetics, doi:10.3390/cosmetics10020047_

Round 1

Reviewer 1 Report

The Author presented a review entitled “Biopigments of Microbial Origin and Their Application in the Cosmetic Industry”.

The review is very interesting, well written and well organized.

The problem of synthetic dyes in cosmetics is carefully highlighted, along searching valid alternatives from natural origins.

Just few suggestion for improving the review before the publication, as follows:

At paragraph “3. Pigments Produced by Microorganisms”, please check and correct that “flavins” is reported twice in the following sentence: “It is well known that filamentous fungi can produce a variety of colors, including carotenoids, flavins, monascins, phenazines, flavins, quinones, and indigo, displaying a large color spectrum”.

In figure 1, please, check the English form of “skin whitening”; in my opinion, it should be better to refer to “skin lightening”.

In the subsection “5.1. Photoprotection and Antioxidant Capacity”, please, check the English form of the following sentence “Antioxidants may also assist in maintaining the organoleptic qualities of cosmetic products by preventing lipid oxidation, which prevents changes in flavor, aroma, and appearance”.

Reviewer 2 Report

Review for

 Biopigments of Microbial Origin and Their Application in the Cosmetic Industry

A literature survey made by a researcher not involved in microbial pigments research

Only one paper published and indexed in Scopus database

Life Science JournalVolume 10, Issue 1, Pages 603 - 608March 2013

Document type

Article

Source type

Journal

ISSN

10978135

View more

Characterization of a novel strain of the genus Actinopolyspora, an extremely halophilic actinomycete isolated from Saudi Arabia

Kiki, Manal J.a

a Department of Biology, King Abdulaziz University, Jeddah, Saudi Arabia

b Department of Biology, Taiba University, Al-Madiah Al-Munawwarh, Saudi Arabia

------------------------------------------------

Classic construction, no true novelty

Covers only part of microbial pigments

Skin whitening in figure 1 not whitenning

-----------------

Interesting part starts with

5. Application of Microbial Pigments in Cosmetics

5.1. Photoprotection and Antioxidant Capacity

5.2. Antiaging Compounds and Skincare Products

5.3. Whitening Effect

5.4. Pigment Substances

------------------------------------------

The author did not even write a single line about the new trend in cosmetics, colorless cosmetics

Colorless Make-Up | Lord & Berry

Lord & Berry

https://www.lordandberry.com › usa

Colorless collection contains essential multitasking makeup products that combine makeup effect with skincare benefits. Colorless collection products ...

-------------------

colorless Lipstick - With Sun Protection ...

GEODERM

https://www.geoderm.com › lipstick

colorless cosmetics    www.geoderm.com

Active Ingredients. • Castor oil: It is a valuable ally of our beauty, thanks to its regenerative power and its calming properties ...

---------------------------

Application of the colorless carotenoids,phytoene,and phytofluene in cosmetics, wellness, nutrition, and therapeutics (  Book Chapter)              von Oppen-Bezalel, L., Shaish, A.              2019      The Alga Dunaliella  pp. 423-444

The decoloring technology of colorless cosmetic oil based on cold-pressed crude camellia oil      Wang, H., Yang, X., Gong, J., (...), Zhang, Y., Tang, J. 2015      Journal of the Chinese Cereals and Oils Association  30(7), pp. 58-63

--------------------------------

Reviewer 3 Report

I read with interest the manuscript ID: cosmetics-2268795 entitled “Biopigments of Microbial Origin and Their Application in the Cosmetic Industry”.

The manuscript is scientifically sound and meets the journal's expectations. It is well written and presented, providing comprehensive and detailed information.

The author made a complex review highlighting the characteristics, the classification, and the application and significance of microbial pigments from prokaryotic and eukaryotic sources and their application in the cosmetic industry.

Current state of the article is very high, the conclusions are properly drawn and also, the provided literature is relevant to the research.

The manuscript has high potential for scientific community and conclude that to assess the effectiveness and safety of employing microbial pigments in the cosmetic industry, further research is still required and new sources of blue pigments need to be discovered, especially in terms of compounds of a natural origin.

Congratulations!

Round 2

Reviewer 2 Report

Revision is OK